# HIERATOK: MULTI-SCALE VISUAL TOKENIZER IMPROVES IMAGE RECONSTRUCTION AND GENERATION

## ABSTRACT

In this work, we present HieraTok, a novel multi-scale Vision Transformer (ViT)-based tokenizer that overcomes the inherent limitation of modeling single-scale representations. This is realized through two key designs: (1) multi-scale downsampling applied to the token map generated by the tokenizer encoder, producing a sequence of multi-scale tokens, and (2) a scale-causal attention mechanism that enables the progressive flow of information from low-resolution global semantic features to high-resolution structural details. Coupling these designs, HieraTok achieves significant improvements in both image reconstruction and generation tasks. Under identical settings, the multi-scale visual tokenizer outperforms its single-scale counterpart by a 27.2% improvement in rFID ($1.47 \rightarrow 1.07$). When integrated into downstream generation frameworks, it achieves a $1.38\times$ faster convergence rate and an 18.9% boost in gFID ($16.4 \rightarrow 13.3$), which may be attributed to the smoother and more uniformly distributed latent space. Furthermore, by scaling up the tokenizer's training, we demonstrate its potential by a sota rFID of 0.45 and a gFID of 1.82 among ViT tokenizers. To the best of our knowledge, we are the first to introduce multi-scale ViT-based tokenizer in image reconstruction and image generation. We hope our findings and designs advance the ViT-based tokenizers in visual generation tasks.

## 1 INTRODUCTION

The rapid advancement of latent generative models (Rombach et al., 2022; Saharia et al., 2022; Fan et al., 2024; Ramesh et al., 2021; 2022) has revolutionized visual content creation, achieving remarkable success in high-quality image synthesis through architectures like diffusion models (Peebles & Xie, 2023; Rombach et al., 2022; Ho et al., 2020; Lipman et al., 2022) and autoregressive transformers (Esser et al., 2021; Vaswani et al., 2017). These tasks typically follow a two-stage training paradigm: first, a visual tokenizer (Bank et al., 2023; Kingma et al., 2013; He et al., 2022) is trained to compress images into latent representations; second, a generative model learns to map conditions to the latent distribution. While current research predominantly focuses on developing novel generative paradigms (Li et al., 2024; Tian et al., 2024) and scaling up generative models (OpenAI, 2025; Google, 2025; Wang et al., 2025), we emphasize that the design of the visual tokenizer is equally crucial. The reconstruction quality of the visual tokenizer fundamentally determines the upper bound of image generation quality, while the regularity of the latent space (Skorokhodov et al., 2025; Yao et al., 2025) governs the convergence speed of downstream generative models.

A key challenge in designing an efficient visual tokenizer is its ability to capture the hierarchical nature of visual information. Natural images inherently exhibit scale consistency, spanning from global semantics to local details. As a result, visual tasks have consistently benifited from multi-scale designs (Ronneberger et al., 2015; Liu et al., 2021; Huang et al., 2024; Fan et al., 2021) , as evidenced by the success of FPN (Lin et al., 2017) in object detection and segmentation, as well as multi-scale autoregressive models like VAR (Tian et al., 2024) and FlowAR (Ren et al., 2024) in generation tasks.

Convolutional architectures naturally model multi-scale representations by progressively reducing spatial dimensions and increasing channel depth. However, for Vision Transformer-based tokenizers (Hansen-Estruch et al., 2025; Xiong et al., 2025; Dosovitskiy et al., 2020), they are inherently limited to modeling single-scale representations, and no multi-scale ViT Tokenizer have been proposed so far. We attribute this limitation to the following challenges:

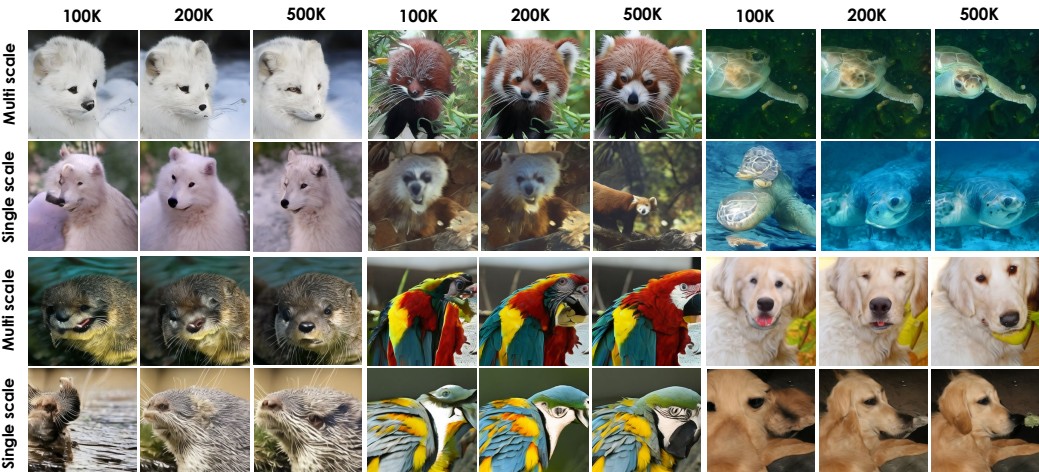

Figure 1: Comparative visualization of generation quality: multi-scale vs. single-scale tokenizers across increasing generative model training steps. (a) Top row: Samples generated by multi-scale tokenizer. (b) Bottom row: Corresponding outputs from the single-scale baseline. We train these models using the DiT-XL framework and conduct inference under identical conditions.

(1) **Fixed-length token sequences**: Convolutional networks adjust feature map sizes through convolution operations. In contrast, for ViT-based models, after patchfying the input image to a token map of a specific resolution, the number of tokens remains fixed throughout the network. This inherently restricts the model to representing fixed-length, single-scale features.

(2) **Global attention mechanism in Transformers**: While the global attention (Vaswani et al., 2017) mechanism in transformers enables comprehensive token interactions, its uniform treatment of all tokens (despite positional encoding) presents difficulties in establishing hierarchical relationships. The self-attention mechanism inherently lacks inductive biases for modeling resolution-dependent feature hierarchies, making it challenging to naturally maintain the progression from high-level semantic representations to low-level structural details that is characteristic of convolutional architectures.

Building on these insights, we introduce HieraTok, a novel and efficient multi-scale tokenizer. To overcome the fixed token sequence length in ViT, we generate a sequence of multi-scale token maps by downsampling the single-scale token map produced by the encoder. During the transformer's processing of these multi-scale token maps, we implement a scale-causal attention mechanism, which restricts high-resolution tokens to interact exclusively with preceding low-resolution tokens. This design ensures a progressive flow of information from coarse-grained semantic tokens to fine-grained structural tokens, effectively establishing a hierarchical transition from global to fine-grained local features. As seen in the Figure 3, this approach mirrors the principles of the FPN (Lin et al., 2017) in convolutional architectures. Furthermore, we apply corresponding interpolation to the ground truth RGB images, enabling each scale's token map to reconstruct the RGB image at its respective scale. Such supervision also embeds scale consistency into the representation space.

Our HieraTok achieves significant improvements in both reconstruction fidelity and generation efficiency. Experimental results validate that under identical training setting,the multi-scale ViT tokenizer significantly outperforms its single-scale counterpart in reconstruction tasks by a 27.2% improvement in rFID ($1.47 \rightarrow 1.07$). In generation tasks, our tokenizer achieves faster convergence rates in diffusion generation frameworks. Additionally, through analyses such as t-SNE (Van der Maaten & Hinton, 2008) in Figure5, we observe that the latent space of the multi-scale ViT tokenizer is smoother and more uniformly distributed. This characteristic likely contributes to the accelerated training of generative models. These experiments confirm the effectiveness of our multi-scale tokenizer design. Furthermore, under a scaled-up experimental setting, HieraTok achieves a sota rFID of 0.45 and a highly competitive gFID of 1.82 among ViT-based tokenizers. We hope this multi-scale approach will drive further breakthroughs in image generation tasks for existing ViT-based tokenizers.

## 2 RELATED WORK

### 2.1 TOKENIZERS FOR IMAGE RECONSTRUCTION

Modern tokenizers compress images into continuous or discrete latent representations. Continuous tokenizers, like AE (Bank et al., 2023; Chen et al., 2024; Michelucci, 2022) and VAE (Kingma et al., 2013; 2019; Yang et al., 2024), encode images into continuous feature spaces or latent distributions. In contrast, discrete tokenizers such as VQ-VAE (Van Den Oord et al., 2017; Razavi et al., 2019; Zheng et al., 2022) employ quantization to obtain discrete codes. During the first stage of generative modeling, tokenizers are optimized for high-fidelity reconstruction, where continuous representations often outperform their discrete counterparts. However, empirical observations (Yu et al., 2023; Hansen-Estruch et al., 2025; Dosovitskiy et al., 2020) reveal an intriguing trade-off: improved reconstruction quality frequently correlates with degraded generation performance. Recent advances (Xiong et al., 2025; Hansen-Estruch et al., 2025) have sought to address this limitation through better tokenizer architectures design. In this work, our multiscale tokenizer preserves structural consistency across scales, jointly improving reconstruction fidelity and generation quality.

### 2.2 IMAGE GENERATION PARADIGM

Generation models learn a mapping from conditions to latent space distributions. Diffusion-based approaches (e.g., LDM (Rombach et al., 2022), DiT (Peebles & Xie, 2023)) operate in continuous spaces (Lipman et al., 2022), while autoregressive transformers (e.g., VQGAN (Esser et al., 2021)) generate discrete tokens. Recently, methods like MAR (Li et al., 2024; Fan et al., 2024) have bridged these paradigms by introducing diffusion losses into autoregressive frameworks, enabling continuous autoregressive generation. Beyond architectural choices, recent work also focus on improving convergence speed. For instance, REPA (Yu et al., 2024b) accelerates generation by aligning intermediate representations with pretrained visual foundation models (Radford et al., 2021; Caron et al., 2021; Oquab et al., 2023). Meanwhile, studies like VAVAE (Yao et al., 2025) and EQ-VAE (Yao et al., 2025) highlight the critical role of latent space structure in governing training difficulty. Our work builds on these insights by proposing a multi-scale latent space design, where explicit scale consistency yields a more generation-friendly representation (Skorokhodov et al., 2025).

### 2.3 MULTI-SCALE VISUAL DESIGN

Scale invariance is a fundamental property of images. Hierarchical multi-scale features—from coarse global structures to fine local details—provide significant benefits for vision tasks. In traditional detection and segmentation, architectures like FPN (Lin et al., 2017), U-Net (Ronneberger et al., 2015), and later Swin-Transformer (Liu et al., 2021), ViTDet (Li et al., 2022), and Mask2Former (Cheng et al., 2022) have demonstrated the critical role of multi-scale design for fine-grained visual understanding. This principle extends to image generation: VAR (Tian et al., 2024) achieved substantial quality improvements in discrete autoregressive generation through multi-scale quantization and autoregressive strategies, while Ming-Lite (Gong et al., 2025) enhanced fine-grained details via learnable multi-scale queries. Our work identifies unique challenges in adapting multi-scale designs to ViT-based tokenizers. We address this with a simple yet effective architecture that significantly improves both reconstruction and generation quality.

## 3 METHOD

### 3.1 PRELIMINARY: VANILLA VIT TOKENIZER

In the vanilla Vision Transformer (ViT) (Dosovitskiy et al., 2020) framework, the tokenizer encodes an input image $\mathbf{X}$ by first applying patch embedding through convolutional or linear layers, yielding a grid of $\frac{h}{p} \times \frac{w}{p} \times d$ tokens, where $h$ and $w$ denote the height and width of the input image, $p$ represents the patch size, and $d$ is the embedding dimension. These tokens are subsequently flattened into a one-dimensional sequence $\mathbf{T}_{\text{flat}} \in \mathbb{R}^{N \times d}$, where $N = \frac{h}{p} \times \frac{w}{p}$, and processed by the transformer encoder $E(\cdot)$ to generate a single-scale representation:

$$\mathbf{Z} = E(\mathbf{X}).\tag{1}$$

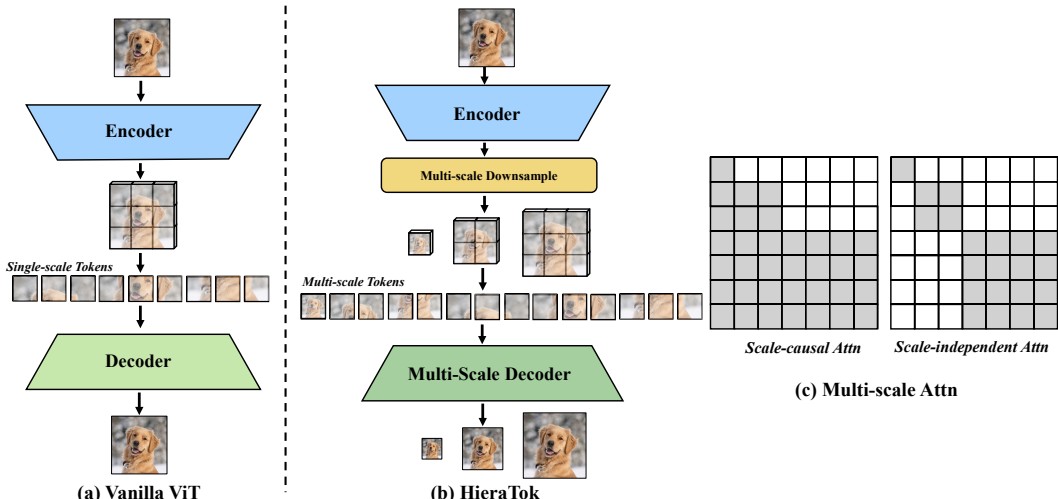

Figure 2: Different tokenizer designs: (a) Single-scale tokenizer ; (b) Our multi-scale tokenizer, featuring two key designs of multi-scale downsampling modules and multi-scale attention mechanisms in the decoder; (c) Multi-scale attention variants: Scale-causal and Scale-independent.

This representation is then passed through a transformer decoder $D(\cdot)$, where each token is projected back to pixel space to reconstruct the complete image:

$$\hat{\mathbf{X}} = D(\mathbf{Z}). \tag{2}$$

For an autoencoder, the training objective is to minimize the discrepancy between the reconstructed image $\hat{\mathbf{X}}$ and the original input image $\mathbf{X}$. This is achieved through a composite loss function:

$$\mathcal{L} = \mathcal{L}_{\text{rec}} + \lambda_1 \mathcal{L}_p + \lambda_2 \mathcal{L}_g + \lambda_3 \mathcal{L}_{\text{kl}}, \tag{3}$$

where $\mathcal{L}_{\text{rec}}$ computes the reconstruction loss, typically using a combination of $L_1$ loss and MSE loss:

$$\mathcal{L}_{\text{rec}} = \beta_1 \mathcal{L}_1 + \beta_2 \mathcal{L}_2. \tag{4}$$

$\mathcal{L}_p$ represents a perceptual loss (Zhang et al., 2018) captureing high-level perceptual differences, and $\mathcal{L}_g$ is a discriminative loss, like the discriminator loss in StyleGAN (Goodfellow et al., 2020; Karras et al., 2019), which enhances the realism of the generated images. The KL divergence loss (Kingma et al., 2013) $\mathcal{L}_{\text{kl}}$ regularizes the latent distribution to approximate a standard normal distribution.

**Limitations of Single-Scale Representation:**  The vanilla ViT tokenizer operates on a single scale, transforming an image into a fixed grid of tokens, which restricts its capacity to capture hierarchical features. In contrast, convolutional tokenizers naturally encode multi-scale information through a fine-to-coarse compression in the encoder and a coarse-to-fine reconstruction in the decoder, preserving both global context and local details. To address the limitations of the vanilla ViT, we propose a multi-scale design that enhances its capability to represent hierarchical features.

### 3.2 MULTI-SCALE VIT TOKENIZER DESIGN

As shown in Figure 2, we propose a multi-scale ViT tokenizer with minimal architectural modifications, addressing two critical aspects: the construction and the interaction of multi-scale representations. To ensure scale consistency, we employ multi-scale RGB supervision during training.

**Multi-Scale Tokens Construction**  To overcome the limitation of fixed token counts in vanilla ViT, we introduce a multi-scale downsampling module $\mathcal{D}(\cdot)$ following the encoder $E(\cdot)$. Given the initial single-scale token map $\mathbf{Z}_0 \in \mathbb{R}^{\frac{H}{P} \times \frac{W}{P} \times d}$ produced by the encoder, we generate a set of token maps $\{\mathbf{Z}_s\}_{s=0}^{S}$ at different scales. Each token map $\mathbf{Z}_s \in \mathbb{R}^{N_s \times d}$ corresponds to a specific scale $s$,

| Tokenizer | Generative Models | rFID ↓ | gFID(w/o CFG) ↓ | gFID(w CFG) ↓ |
|---|---|---|---|---|
| *Conv-based Tokenizer* | | | | |
| LDM(Rombach et al., 2022) | MAR(Li et al., 2024) | 0.53 | 2.35 | 1.55 |
| SD-VAE(sd-, 2023) | REPA(Yu et al., 2024b) | 0.61 | 5.90 | 1.42 |
| SD-VAE | MDT(Reuss et al., 2024) | 0.61 | 6.23 | 1.79 |
| SD-VAE | MDTv2(Gao et al., 2023) | 0.61 | - | 1.58 |
| SD-VAE | DiT(Peebles & Xie, 2023) | 0.61 | 9.62 | 2.27 |
| SD-VAE | SiT(Ma et al., 2024) | 0.61 | 8.61 | 2.06 |
| SD-VAE | MaskDiT(Zheng et al., 2023) | 0.61 | 5.69 | 2.28 |
| VAR-VAE(Tian et al., 2024) | VAR-d30(Tian et al., 2024) | 1.00 | - | 1.92 |
| VA-VAE(Yao et al., 2025) | LightningDiT(Yao et al., 2025) | 0.28 | 2.17 | 1.35 |
| *ViT-based Tokenizer* | | | | |
| MaskGIT(Chang et al., 2022) | MaskGIT(Chang et al., 2022) | 2.28 | 6.18 | - |
| VQGAN(Esser et al., 2021) | LlamaGen(Sun et al., 2024) | 0.59 | 9.38 | 2.18 |
| FlexTok d18-d28(Bachmann et al., 2025) | LlamaGen(Sun et al., 2024) | 1.45 | - | 1.86 |
| ViTok S-L(Hansen-Estruch et al., 2025) | LlamaGen(Sun et al., 2024) | 0.46 | - | 2.45 |
| TiTok-S(Yu et al., 2024a) | MaskGIT(Chang et al., 2022) | 1.71 | - | 1.97 |
| MAETok(Chen et al., 2025) | LightningDiT(Yao et al., 2025) | 0.48 | - | 1.73 |
| GigaTok-XL-XXL(Xiong et al., 2025) | LlamaGen-XXL(Sun et al., 2024) | 0.79 | - | 1.98 |
| HieraTok | DiT | 0.45 | 3.53 | 1.82 |

Table 1: Performance of HieraTok in the Context of sota Generative Models on ImageNet 256x256. HieraTok achieves a competitive rFID of 0.45 and gFID of 1.82 for ViT-based tokenizers.

where $N_s = \frac{H}{p \cdot 2^{S-s}} \times \frac{W}{p \cdot 2^{S-s}}$. These token maps are then concatenated sequentially from **low to high resolution** to form a multi-scale representation:

$$\mathbf{Z}_{\text{multi}} = \text{Concat}(\mathbf{Z}_0, \mathbf{Z}_1, \ldots, \mathbf{Z}_S) \in \mathbb{R}^{(\sum_{s=0}^{S} N_s) \times d}. \tag{5}$$

For the downsampling module $\mathcal{D}(\cdot)$, we explore two strategies: (1) **interpolation-based downsampling**, where $\mathbf{Z}_s$ is obtained by interpolating $\mathbf{Z}_S$ using area-pooling methods, and (2) **learnable convolutional downsampling**, where $\mathbf{Z}_s$ is generated by applying a trainable convolutional layer $\text{Conv}_s(\cdot)$ to $\mathbf{Z}_S$. Both downsampling methods demonstrate performance gains in reconstruction tasks. Nevertheless, for generation tasks, only the convolution-based downsampling approach exhibits the capability to improve generation quality, while interpolation-based downsampling method does not yield such benefits. A detailed ablation study in Section 5.3 further analyzes these trade-offs.

**Multi-Scale Information Interaction** In the decoder, we design two attention mechanisms to facilitate interaction among multi-scale token representations. The first, termed **scale-independent attention**, restricts each token to interact only with tokens within the same scale, ensuring independence between representations at different resolutions. Mathematically, this can be expressed as:

$$\text{Attention}(\mathbf{Q}_s, \mathbf{K}_s, \mathbf{V}_s) = \text{Softmax}\left(\frac{\mathbf{Q}_s \mathbf{K}_s^\top}{\sqrt{d}}\right) \mathbf{V}_s, \tag{6}$$

where $\mathbf{Q}_s$, $\mathbf{K}_s$, and $\mathbf{V}_s$ denote the query, key, and value matrices for the $s$-th scale, respectively. The second mechanism, **scale-causal attention**, allows tokens at the current resolution to interact with tokens at both the current and

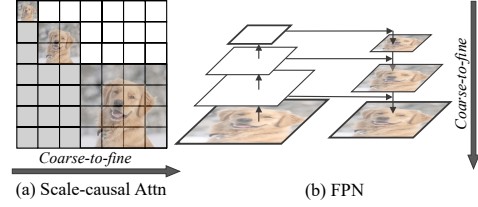

(a) Scale-causal Attn    (b) FPN

Figure 3: Our scale-causal attention operates akin to FPN along the token sequence dimension.

preceding resolutions. Combined with our token concatenation strategy, which proceeds from low to high resolution, this design enables a hierarchical information flow, where high-level semantic features from lower resolutions progressively refine low-level structural details at higher resolutions. Mathematically, this is formulated as:

$$\text{Attention}(\mathbf{Q}_s, \mathbf{K}_{\leq s}, \mathbf{V}_{\leq s}) = \text{Softmax}\left(\frac{\mathbf{Q}_s \mathbf{K}_{\leq s}^\top}{\sqrt{d}}\right) \mathbf{V}_{\leq s}, \tag{7}$$

where $\mathbf{K}_{\leq s}$ and $\mathbf{V}_{\leq s}$ represent the key and value matrices for all resolutions up to and including the $s$-th scale. This information propagation mechanism is analogous to the top-down feature aggregation in Feature Pyramid Networks (FPNs) as shown in Figure 3. By integrating this multi-scale information flow into the transformer's token interactions, each attention operation simultaneously performs a function similar to FPN's feature aggregation along the token sequence dimension.

| Spec. | KL Weight | Tokenizer | Reconstruction | | | | Generation | |
|---|---|---|---|---|---|---|---|---|
| | | | rFID↓ | PSNR↑ | LPIPS↓ | SSIM↑ | gFID↓ | IS↑ |
| f16d16 | 1e-6 | Single Scale | 1.47 | 24.4 | 0.108 | 0.74 | 16.4 | 57.4 |
| | | Multi Scale | 1.07 *(-0.40)* | 23.8 | 0.087 | 0.72 | 13.3 *(-3.1)* | 68.5 *(+11.1)* |
| f16d16 | 1e-5 | Single Scale | 1.76 | 23.9 | 0.122 | 0.72 | 20.2 | 51.0 |
| | | Multi Scale | 1.61 *(-0.16)* | 22.8 | 0.115 | 0.67 | 17.9 *(-2.3)* | 55.5 *(+4.5)* |
| f16d32 | 1e-5 | Single Scale | 1.60 | 24.3 | 0.119 | 0.73 | 24.3 | 43.6 |
| | | Multi Scale | 1.54 *(-0.06)* | 23.0 | 0.113 | 0.68 | 19.4 *(-4.9)* | 52.5 *(+8.9)* |

Table 2: Comparative evaluation of multi-scale versus single-scale tokenizers under various settings reveals the multi-scale variant's consistent performance advantages in both image reconstruction and generation tasks. We highlight the baseline's rFID and gFID in blue for better distinction.

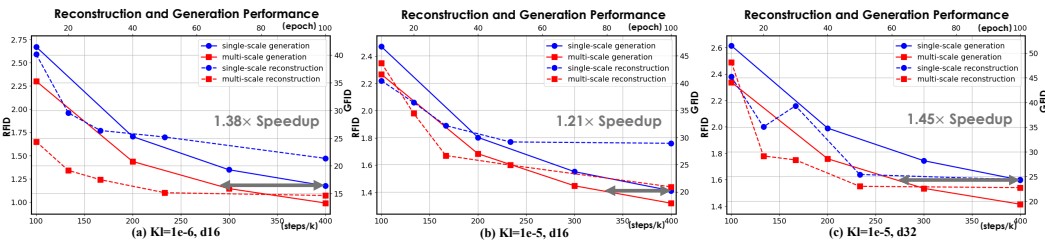

Figure 4: Performance evolution of multi-scale versus single-scale tokenizers across training iterations for both reconstruction and generation tasks. The multi-scale tokenizer demonstrates consistent superiority over its single-scale counterpart throughout the training process.

**Multi-Scale RGB Supervision**   During decoding, multi-scale tokens, like their single-scale counterparts, are regressed to reconstruct the RGB pixels of their corresponding patches. Specifically, we downsample the ground truth image $\mathbf{X}$ to match each scale $s$ using an interpolation operation $\mathcal{D}_s(\cdot)$, and supervise the reconstruction of the corresponding resolution. This supervision effectively introduces **scale equivariance** into the representation space and the decoder. Formally, let $\mathcal{D}_s(\cdot)$ denote the downsampling operation for scale $s$. We establish the following optimization objectives for reconstruction:

$$\mathcal{L}_{\text{full}} = \|D(\mathbf{Z}) - \mathbf{X}\|_2^2, \quad \text{where} \quad \mathbf{Z} = E(\mathbf{X}), \tag{8}$$

$$\mathcal{L}_{\text{scale}} = \|D(\mathcal{D}_s(\mathbf{Z})) - \mathcal{D}_s(\mathbf{X})\|_2^2. \tag{9}$$

where $E(\cdot)$ is the encoder and $D(\cdot)$ is the decoder. This supervision ensures that downsampling operations commute with the latent representation and decoder (i.e., $D(\mathcal{D}_s(\mathbf{Z})) \approx \mathcal{D}_s(D(\mathbf{Z}))$), thereby enforcing scale consistency across resolutions.

We validate that the multi-scale supervised tokenizer achieves significant improvements in both reconstruction and generation performance, while the introduced scale consistency likely leads to better latent space structure and more uniform distributions.

## 4 IMPLEMENTATION DETAILS

**Tokenizer Training**   Following the conclusions from (Hansen-Estruch et al., 2025), we adopt a small ViT encoder and a large ViT decoder for all tokenizer architectures. All tokenizers are trained on the ImageNet 256×256 dataset (Russakovsky et al., 2015) with a learning rate of $1 \times 10^{-4}$ and a global batch size of 256. We evaluate two compression ratios: **f16d16** and **f16d32**, where **f** represents the spatial compression ratio and **d** denotes the dimension of latent channels. For the multi-scale tokenizer, the default multi-scale design is implemented with scales [1,2,4,8,16]. For our ablation studies, we train models for 100 epochs using only L1, L2, perceptual, and KL losses to obtain results quickly. For the scaled-up tokenizer experiments, we supplement these with GAN loss and vf-loss and increase the number of training epochs to achieve better performance.

| Tokenizer | Downsample | Attention | Reconstruction | | | | Generation | |
|-----------|-----------|-----------|------|------|-------|------|------|------|
| | | | rFID↓ | PSNR↑ | LPIPS↓ | SSIM↑ | gFID↓ | IS↑ |
| Single-Scale | - | - | 1.47 | 24.4 | 0.108 | 0.74 | 16.4 | 57.4 |
| Multi-Scale | Conv | Scale-causal | **1.07** | 23.8 | 0.087 | 0.72 | **13.3** | 68.5 |
| | Conv | Scale-independent | 1.13 | 23.7 | 0.093 | 0.71 | 14.0 | 65.3 |
| | Conv | Full-Attn | 1.24 | 23.6 | 0.099 | 0.71 | 16.6 | 56.4 |
| | Interpolate | Scale-causal | 1.18 | 23.7 | 0.095 | 0.71 | 17.8 | 52.1 |
| | Interpolate | Scale-independent | 1.16 | 23.7 | 0.091 | 0.71 | 17.2 | 54.4 |
| | Interpolate | Full-Attn | 1.13 | 23.0 | 0.113 | 0.68 | 16.9 | 55.7 |

Table 3: Performance comparison of multi-scale tokenizer variants with different downsampling and attention designs.

| Tokenizer | Scales | Reconstruction | | | | Generation | |
|-----------|--------|------|------|-------|------|------|------|
| | | rFID↓ | PSNR↑ | LPIPS↓ | SSIM↑ | gFID↓ | IS↑ |
| Single-Scale | [16] | 1.47 | 24.4 | 0.108 | 0.74 | 16.4 | 57.4 |
| Multi-Scale | [1,2,4,16] | 1.25 | 23.6 | 0.092 | 0.71 | 15.9 | 60.2 |
| | [1,2,4,8,16] | 1.07 | 23.8 | 0.087 | 0.72 | **13.3** | 68.5 |
| | [1,2,4,8,12,16] | **1.04** | 23.9 | 0.087 | 0.72 | 13.8 | 66.8 |

Table 4: Performance comparison of multi-scale tokenizer variants with different multi-scale setting.

**Performance Evaluation**   We assess the reconstruction quality of the tokenizer using metrics including FID (Heusel et al., 2017), PSNR (Hore & Ziou, 2010), and SSIM (Wang et al., 2004) on the ImageNet-50k validation dataset. To validate the performance of different tokenizers on generative tasks, we conduct extensive experiments using the **vanilla DiT** framework as our testbed. Specifically, we employ DiT-XL as our default generative model with a global batch size of 256 and a constant learning rate of 1e-4 without decay. For generative performance evaluation, we generate 50k images without classifier-free guidance (CFG) (Ho & Salimans, 2022) and compute both gFID and IS (Barratt & Sharma, 2018) metrics. For our ablation studies, all tokenizer variants are trained on DiT-XL (Peebles & Xie, 2023) for **400k steps**. For the scaled-up experiments, we train for **5M** steps to achieve superior generation performance.

## 5 EXPERIMENTS RESULTS AND ANALYSIS

### 5.1 MULTI-SCALE TOKENIZER IMPROVES GENERATION AND RECONSTRUCTION

Our results demonstrate that the multi-scale tokenizer consistently improves both reconstruction and generation performance over its single-scale counterpart across different KL divergence weights (1e-5 and 1e-6) and latent compression ratios (f16d16 and f16d32) settings as shown in Figure 4 and Table 2. For reconstruction, under the KL=1e-6, d16 compression setting, HieraTok achieves a 27.2% reduction in rFID and a 19.4% improvement in LPIPS (Table 2), highlighting its superior perceptual fidelity. For generation, under strictly controlled training settings (400k steps), the multi-scale tokenizer not only converges 1.38x faster but also yields a significant 18.9% gFID improvement (16.4 → 13.3), as shown in Figure 4. We further validate this advantage across various DiT model sizes (Table 6), confirming the robustness of our multi-scale design.

To situate HieraTok within the state-of-the-art landscape, we also conduct a scaled-up experiment using an enhanced training strategy over 5M steps. The resulting model achieves a new sota rFID of 0.45 among ViT-based tokenizers and a highly competitive gFID of 1.82 (Table 1). This result underscores the high performance ceiling of our proposed architecture, complementing the direct, controlled comparisons that validate the effectiveness of the core design.

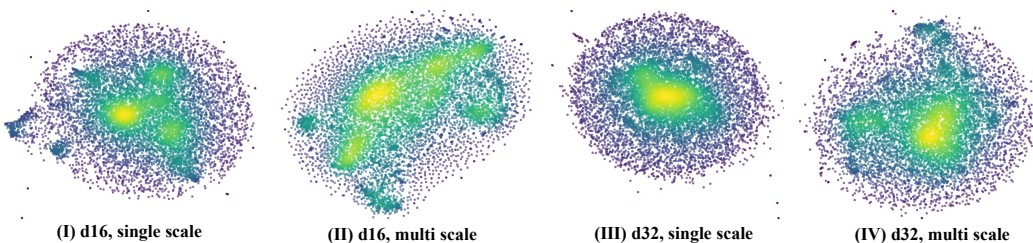

(I) d16, single scale     (II) d16, multi scale     (III) d32, single scale     (IV) d32, multi scale

Figure 5: **t-SNE visualization of latent space representations**, demonstrating that the multi-scale tokenizer achieves more uniform feature distribution compared to the single-scale baseline.

| Spec. | Tokenizer | Density CV↓ | Gini Coefficient↓ | Normalized Entropy↑ | gFID↓ |
|---|---|---|---|---|---|
| f16d16 | Single-Scale | 0.522 | 0.299 | 0.984 | 16.4 |
| | Multi-Scale | 0.381 | 0.216 | 0.991 | 13.3 |
| f16d32 | Single-Scale | 0.656 | 0.371 | 0.976 | 24.3 |
| | Multi-Scale | 0.435 | 0.249 | 0.988 | 19.4 |

Table 5: **Analysis of latent space uniformity**, demonstrating that the multi-scale design achieves more uniform latent distributions while simultaneously improving generation quality.

## 5.2 VISUALIZATION AND ANALYSIS OF LATENT SPACE STRUCTURE

To investigate the intrinsic advantages of our multi-scale design, we conduct a comprehensive latent space analysis using t-SNE (Van der Maaten & Hinton, 2008) visualization in Figure 5. We analyze the latent distributions of 10k ImageNet test images using t-SNE projections in Table 5, following VAVAE's (Yao et al., 2025) methodology by calculating the standard deviation and Gini coefficients through kernel density estimation. Both qualitative visualizations and quantitative metrics demonstrate that the multi-scale tokenizer achieves more uniform latent space distributions. Notably, we observe that increasing the tokenizer's latent dimension leads to over-concentrated clusters, potentially explaining the training difficulties in high-dimensional settings, while our multi-scale architecture consistently alleviates this issue. These findings collectively suggest that the multi-scale tokenizer's superior performance, particularly its faster convergence in generation tasks, stems from its better-structured latent geometry and more homogeneous distribution properties.

## 5.3 ANALYSIS ON MULTI-SCALE DOWNSAMPLE METHODS

As shown in Table 3, we investigated two distinct downsampling strategies: a parameter-free approach using direct interpolation and a learnable approach based on a series of convolutional kernels. Both strategies led to substantial gains in reconstruction fidelity. However, we observed a critical performance trade-off when applying these models to generative tasks. The learnable convolutional method consistently improved generation quality, while the direct interpolation strategy's performance was not only significantly inferior but also failed to surpass the single-scale generation baseline.

We hypothesize that the divergent outcomes stem from the fundamental differences between the two tasks. In reconstruction, the model benefits directly from multi-scale supervision, where each level of the decoder is guided by a corresponding ground-truth image scale. This multi-level guidance is the primary source of improvement, placing less stringent demands on the latent representation itself. Generation, however, is critically dependent on the learned latent distribution. We postulate that a simple interpolation-based downsampler cannot effectively differentiate between high-level semantic content and low-level visual textures. This makes it difficult to form a well-structured, hierarchical representation with distinct levels of feature granularity. Therefore, learnable convolutional kernels are essential, as they actively learn transformations that separate and specialize the visual information appropriate for each scale.

| Generation Model | Type | Params | Tokenizer | Generation Performance | |
|---|---|---|---|---|---|
| | | | | gFID↓ | IS↑ |
| DiT-XL | Diff | 675M | Single Scale | 16.4 | 57.4 |
| | | | Multi Scale | 13.3 *(-3.1)* | 68.5 *(+11.1)* |
| DiT-L | Diff | 458M | Single Scale | 19.3 | 50.7 |
| | | | Multi Scale | 16.5 *(-2.8)* | 58.9 *(+8.2)* |
| DiT-B | Diff | 130M | Single Scale | 33.6 | 31.8 |
| | | | Multi Scale | 30.8 *(-2.8)* | 35.2 *(+3.4)* |

Table 6: **Generation performance evaluation across different DiT model sizes**, demonstrating consistent superiority of the multi-scale tokenizer over the single-scale baseline.

### 5.4 ANALYSIS ON MULTI-SCALE ATTENTION MECHANISM

We conducted an ablation study on three different attention mechanisms under the KL=1e-6, d16 setting: (1) the standard full-attention from the original ViT; (2) our proposed scale-causal attention; and (3) a scale-independent variant. The results in Table 3 show that while all three mechanisms improve reconstruction quality, their generative performance varies considerably. The scale-causal attention markedly outperforms both the full-attention and scale-independent mechanisms in generation tasks.

We attribute the reconstruction improvements to the inherent benefits of multi-scale representations and supervision, as discussed in the previous Section 5.3. Generative modeling, however, imposes stricter requirements on the structure of the latent representations. Although full-attention allows for bidirectional information flow between coarse and fine scales, its unrestricted visibility makes it difficult to differentiate between scale-specific features. This limits its ability to effectively model the coarse-to-fine progression from global semantics to local details, resulting in generative performance comparable to a single-scale tokenizer. Conversely, the scale-independent mechanism successfully isolates representations at each scale but lacks the inter-scale interaction needed to fully leverage the multi-scale advantage. Only the scale-causal design, which models a progressive refinement process analogous to FPN where low-resolution semantic features inform high-resolution structural features, effectively optimizes the latent space structure to yield substantial gains in generation quality.

### 5.5 ANALYSIS ON MULTI-SCALE SETTINGS

Through systematic investigation of multi-scale configurations under the kl=1e-6 and f16d16 parameter settings, we evaluate three distinct scale combinations and assess their impact on both reconstruction and generation performance in Table 4. Our experiments demonstrate that incorporating just three scales (1, 2, 4), corresponding to 21 tokens, yields a significant 15% improvement in reconstruction quality (from 1.47 to 1.25), with further scale augmentation consistently enhancing reconstruction performance. For generation tasks, while multi-scaling improves results, we observe diminishing returns beyond five scales (1, 2, 4, 8, 16), suggesting this combination may represent the performance ceiling for this approach. Based on these empirical findings, we recommend the (1, 2, 4, 8, 16) configuration as optimal for balancing computational efficiency with performance gains.

## 6 CONCLUSION

In this work, we introduce a novel multi-scale ViT tokenizer HieraTok that overcomes the limitation of vanilla ViT in modeling only single-scale token maps. Through meticulous design of multi-scale tokens construction and inter-scale information interaction mechanisms, we demonstrate that our multi-scale tokenizer significantly outperforms its single-scale counterpart in both reconstruction and generation tasks. Furthermore, our investigation reveals that multi-scale representation and supervision introduce scale consistency, which regularizes the latent space structure, resulting in a smoother and more uniform latent space distribution. This property may explain why the multi-scale design accelerates convergence in generation tasks. Collectively, these findings strongly suggest that our proposed multi-scale ViT tokenizer represents a substantial advancement over existing ViT-based tokenizer architectures. We believe this method holds great promise as a standard strategy for training high-performance tokenizers for image reconstruction and generation.

# 7 ETHICS STATEMENT

This research does not raise any ethical concerns.

# 8 REPRODUCIBILITY STATEMENT

The appendix provides detailed architecture and training parameters for tokenizer in Appendix E and training parameters for vit in Appendix F.

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

## A  USE OF LLM

**The use of a Large Language Model (LLM) is limited to language editing and refinement.**

## B  DERIVATION OF THE DECODER-DOWNSAMPLING APPROXIMATION

This section provides a detailed derivation for the approximation $D(D_s(Z)) \approx D_s(D(Z))$. This relationship, which suggests that the decoder and downsampling operations are approximately commutative, is foundational to our multi-scale supervision strategy. The validity of this approximation is contingent on a well-trained tokenizer capable of high-fidelity image reconstruction.

Let $X$ represent a high-resolution image, and let $Z = E(X)$ be its corresponding latent representation obtained from the encoder $E(\cdot)$. Let $D(\cdot)$ be the decoder, and let $D_s(\cdot)$ be the downsampling operation for a given scale $s$. The derivation proceeds as follows:

1. **Multi-Scale Reconstruction Objective:** The multi-scale supervision loss, $\mathcal{L}_{\text{scale}}$, is designed to train the decoder to reconstruct a downsampled image, $D_s(X)$, from a correspondingly downsampled latent code, $D_s(Z)$. For an optimally trained model, this objective leads to the following approximation:

$$D(D_s(Z)) \approx D_s(X) \qquad (10)$$

2. **Full-Resolution Reconstruction Objective:** Simultaneously, the primary reconstruction objective trains the decoder to reconstruct the original full-resolution image $X$ from the complete latent code $Z$. For a well-trained model, this yields:

$$D(Z) \approx X \qquad (11)$$

3. **Downsampling the Full Reconstruction:** By applying the downsampling operator $D_s(\cdot)$ to both sides of the approximation in Equation equation 11, we obtain:

$$D_s(D(Z)) \approx D_s(X) \qquad (12)$$

4. **Establishing Equivalence:** The expressions in Equation equation 10 and Equation equation 12 are both approximately equal to the same term, $D_s(X)$. By equating their left-hand sides, we arrive at the final derived relationship:

$$D(D_s(Z)) \approx D_s(D(Z)) \qquad (13)$$

This derivation formally demonstrates that if a decoder can accurately reconstruct images at both full and reduced scales from their respective latent codes, the operations of decoding and downsampling become interchangeable in practice.

## C  ANALYSIS OF COMPUTATIONAL COST AND TRAINING FAIRNESS

A detailed comparison of the computational costs and the fairness of the training protocol is provided to ensure full transparency. A distinction is made between the two primary training stages: tokenizer training (reconstruction) and downstream model training (generation).

### C.1  COMPUTATIONAL COST

Table 7 presents a comparison of the key computational metrics for the single-scale and multi-scale tokenizers.

During the **tokenizer training (reconstruction) stage**, the multi-scale model exhibits a higher computational cost. This is primarily attributed to its decoder, which processes a longer sequence of concatenated tokens from different scales.

However, for the **downstream model training (generation) stage**, the methodology is designed to introduce **zero additional computational overhead**. For the generation task, the latent representation is extracted from the encoder *before* the multi-scale downsampling is applied. This ensures that the latent code passed to the subsequent generative model (e.g., a DiT) has the exact same dimensions as the code from the single-scale baseline. Consequently, both the training and inference processes of the downstream generation task remain unaffected in terms of computational cost and token count.

| Tokenizer | Params | GFLOPS | Encoder Tokens | Decoder Tokens |
|---|---|---|---|---|
| single scale | 346.25M | 182.36 | 256 | 256 |
| multi scale | 401.18M | 230.45 | 256 | 341 |

Table 7: Comparison of computational cost for single-scale and multi-scale tokenizers during the reconstruction training phase.

## C.2 TRAINING FAIRNESS

To ensure a fair comparison, the training setups for both tokenizer architectures were closely aligned. Both the single-scale and multi-scale tokenizers utilize identical encoder architectures. Furthermore, both encoders process the same set of images at the same resolution. By training both models for the same number of epochs, we guarantee that each architecture is exposed to the exact same volume and nature of primary visual data from the encoder's perspective. Given this controlled setup, comparing the models based on training epochs is a fair and standard method to evaluate the effectiveness of each tokenizer's architecture in learning to reconstruct and represent visual information.

## D SCALE-CONSISTENT GENERATION

Our HieraTok is trained with multi-scale RGB images as supervision signals, enabling it to generate multi-scale RGB outputs. While the main text only visualizes the largest-scale generated images, we provide additional visualizations in the appendix—showing multi-scale generations using both DiT-XL as the generative model and HieraTok's multi-scale decoder. The consistent generation quality across scales demonstrates that our proposed HieraTok achieves strong scale coherence.

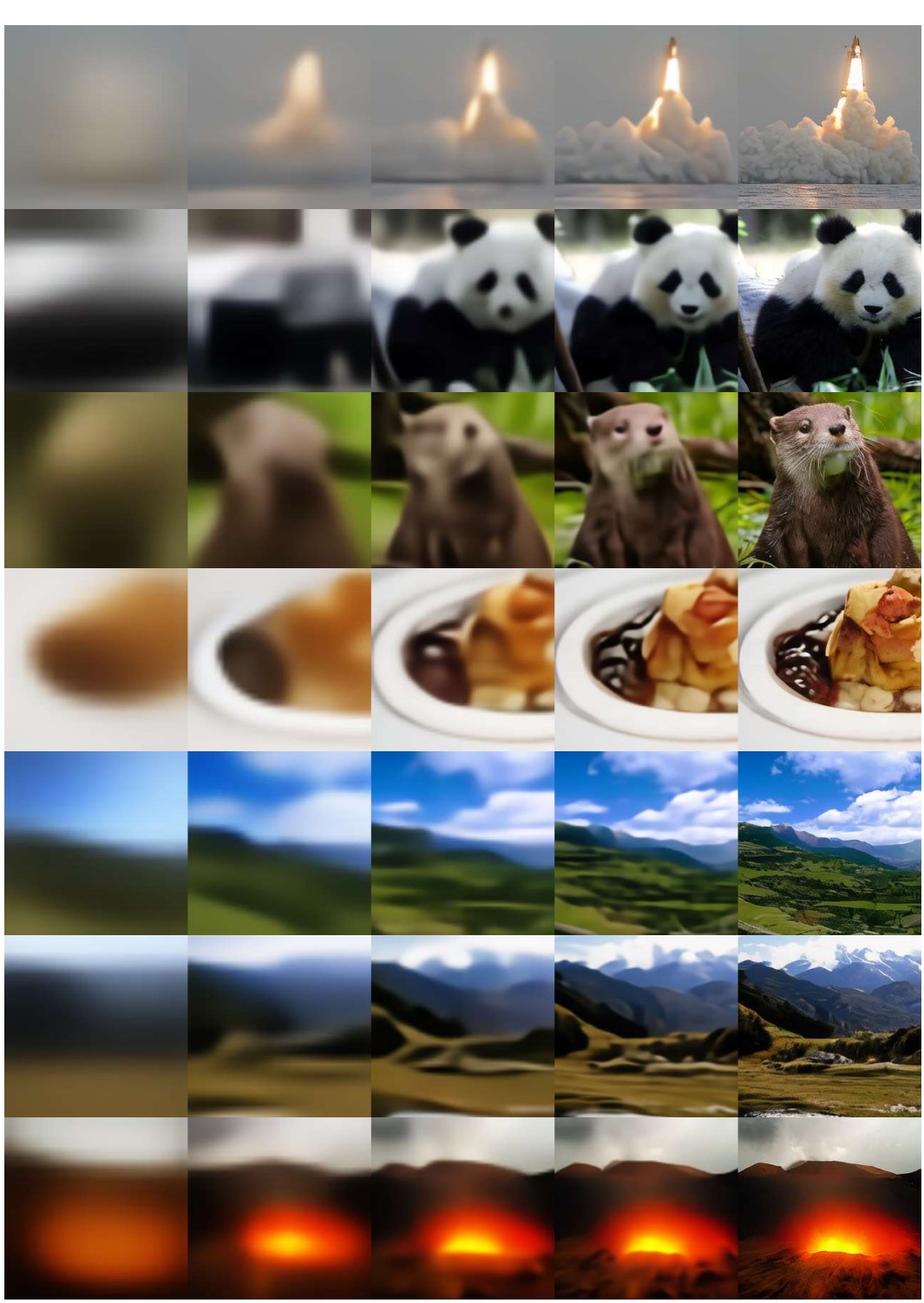

Figure 6: Multi-scale generated samples. Class label 812, 388, 360, 928, 979 and 980.

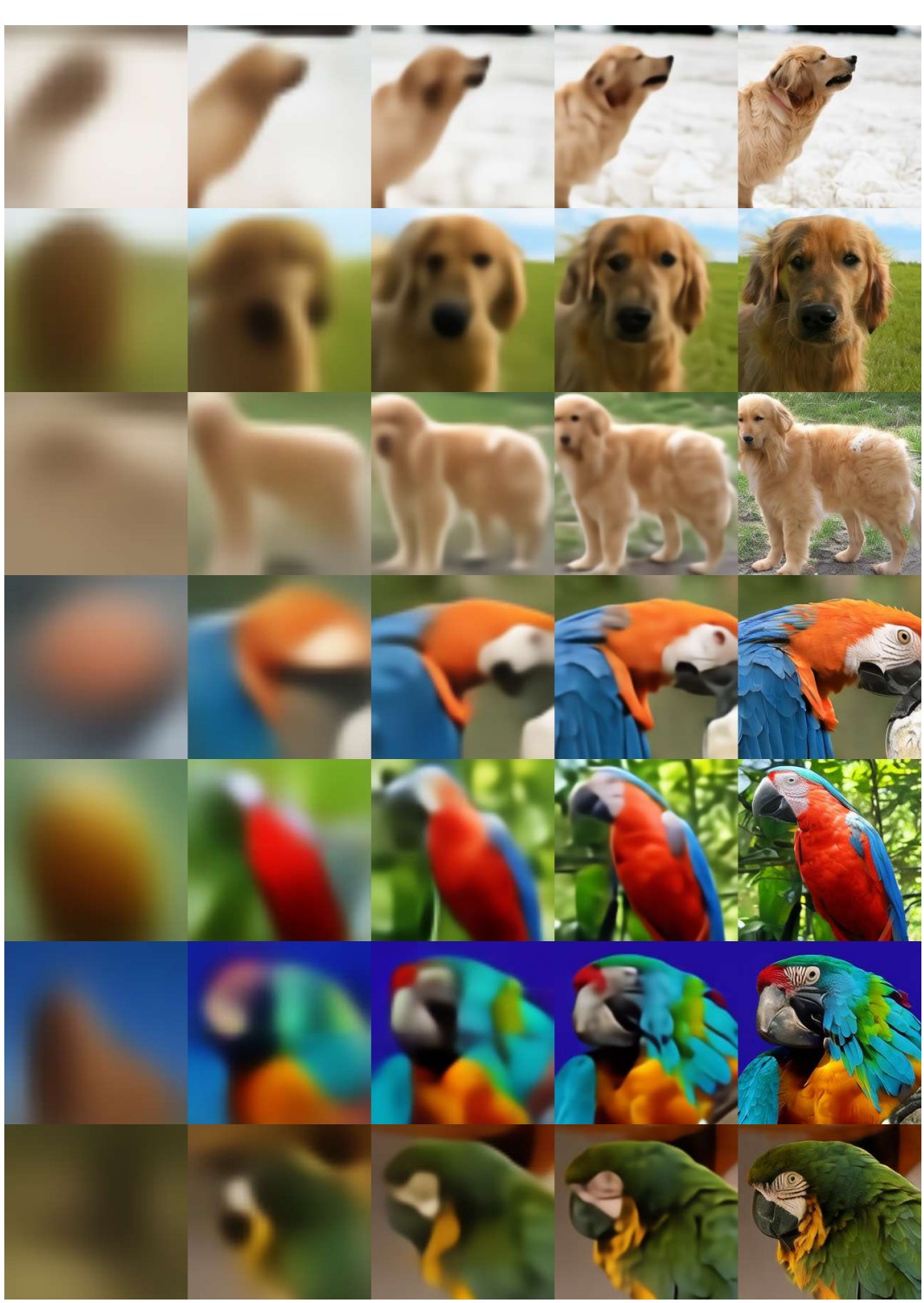

Figure 7: Multi-scale generated samples. Class label 207 and 88.

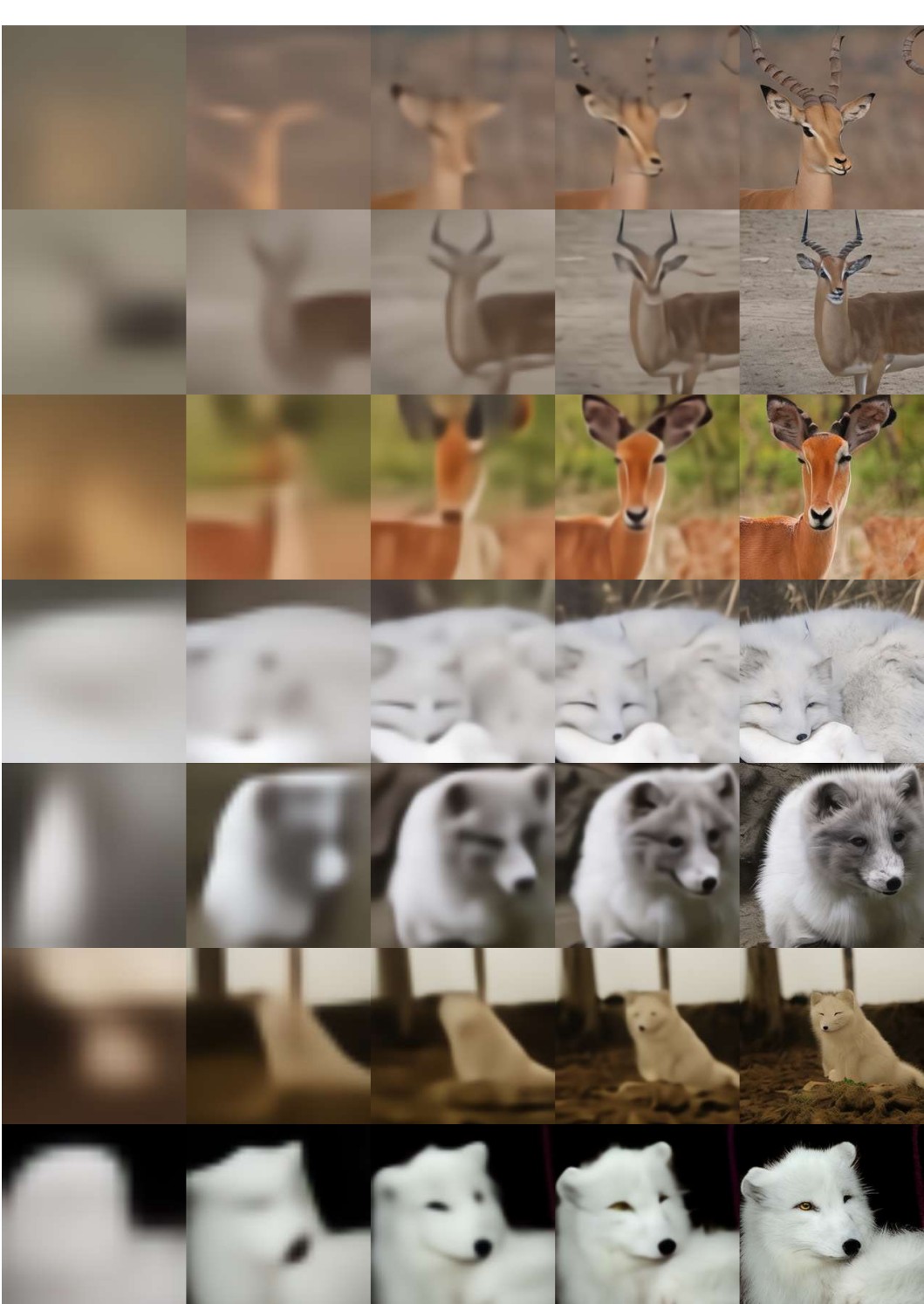

Figure 8: Multi-scale generated samples. Class label 352 and 279.

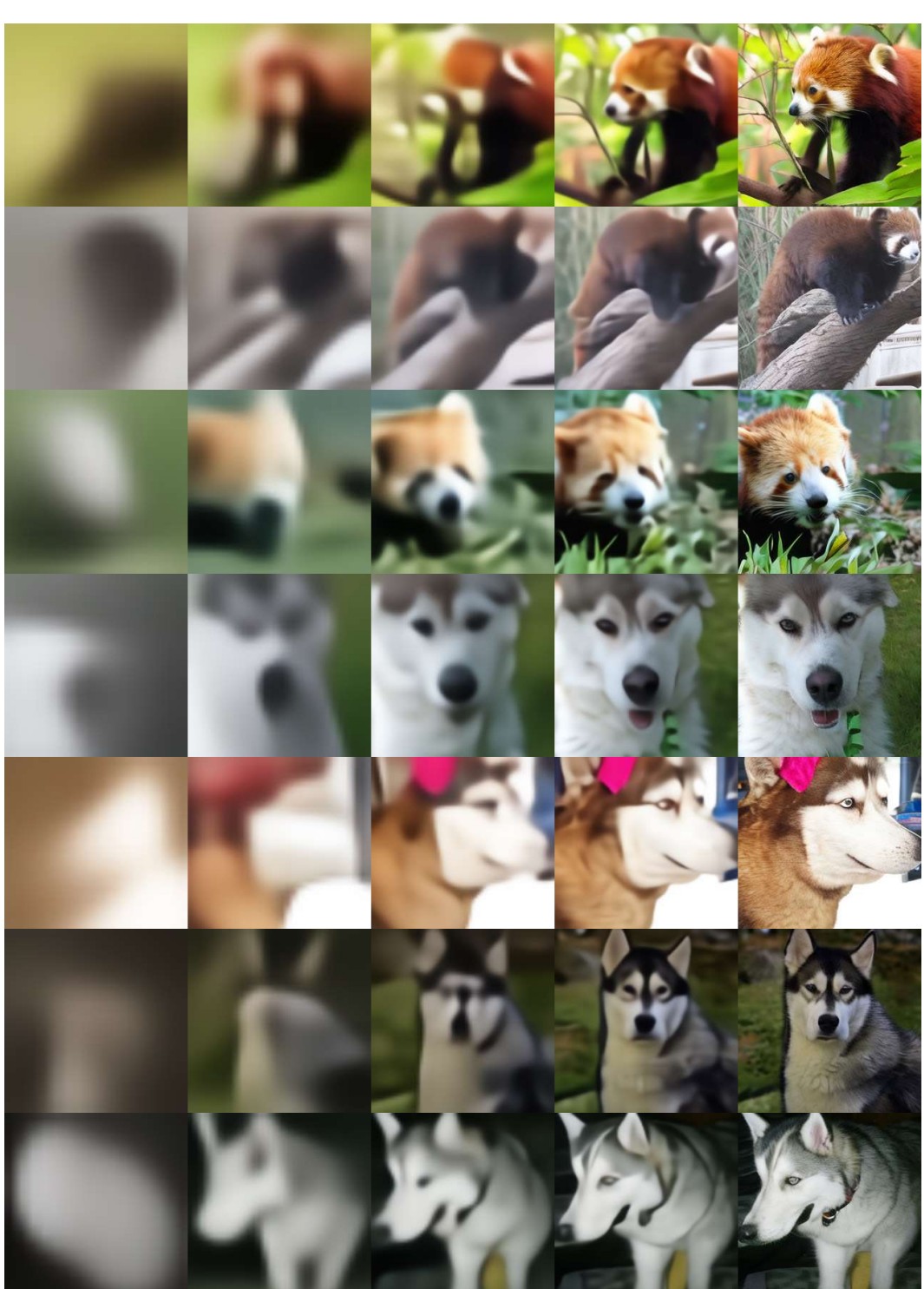

Figure 9: Multi-scale generated samples. Class label 387 and 250.

# E  TOKENIZER IMPLEMENTATION DETAILS

| Parameter | Stage1 | Stage2 |
|---|---|---|
| Traing data | ImageNet | ImageNet |
| Training epochs | 100 | 40 |
| Global Batch size | 256 | 256 |
| Warmup ratio | 0.03 | 0.03 |
| Optimizer | AdamW | AdamW |
| Learning Rate start | $4.0 \times 10^{-4}$ | $2.0 \times 10^{-4}$ |
| Learning Rate end | $4.0 \times 10^{-6}$ | $2.0 \times 10^{-6}$ |
| LR scheduler | CosineAnnealing | CosineAnnealing |
| Optimizer beta1 | 0.9 | 0.9 |
| Optimizer beta2 | 0.95 | 0.95 |
| Weight decay | 0.05 | 0.05 |
| GAN Optimizer | - | AdamW |
| GAN LR start | - | $2.0 \times 10^{-4}$ |
| GAN LR end | - | $2.0 \times 10^{-6}$ |
| GAN LR scheduler | - | CosineAnnealing |
| GAN Warmup ratio | - | 0.01 |
| GAN Optimizer beta1 | - | 0.9 |
| GAN Optimizer beta2 | - | 0.95 |
| GAN Weight decay | - | $1.0 \times 10^{-4}$ |

Table 8: Training settings for Tokenizer.

| Loss Type | Stage1 | Stage1 |
|---|---|---|
| L1 Loss | 1.0 | 1.0 |
| MSE Loss | 0.4 | 0.4 |
| LPIPS Loss | 1.0 | 1.0 |
| KL Loss | 1e-6 | 1e-6 |
| GAN Loss | - | 0.6 |

Table 9: Different Loss Weight.

Our tokenizer adopts the ViT architecture with specific model parameters as shown in the Table 10. We process images into tokens using a convolutional kernel with patch size $p = 16$.

For positional encoding, we employ learnable absolute positional encoding without using rotary positional encoding. To handle multi-scale features at resolutions $s \in \{1, 2, 4, 8, 16\}$, we design $h \times w$ positional encodings $PE_{\text{spatial}} \in \mathbb{R}^{H \times W \times d}$, where $H$ and $W$ are the maximum height and width across all scales and 5 additional scale-specific encodings $PE_{\text{scale}} \in \mathbb{R}^{5 \times d}$. For tokens at resolution $s$, we first interpolate the spatial positional encoding $PE_{\text{spatial}}$ to target size $(H/s, W/s)$ using area-pooling interpolation, then add the corresponding scale encoding $PE_{\text{scale}}^{(s)} \in \mathbb{R}^d$, resulting in the final positional encoding

$$PE^{(s)} = \text{Interpolate}(PE_{\text{spatial}}, (H/s, W/s)) + PE_{\text{scale}}^{(s)}.$$

Similar to ViTDet's design, in the multi-scale downsampling module of the convolution, we process each resolution $s$ using three convolutional kernels, downsampling the token map from the encoder's maximum resolution to the corresponding target resolution.

|  | Encoder | Decoder |
| --- | --- | --- |
| Num Layers | 6 | 24 |
| Hidden Size | 768 | 1024 |
| Num Heads | 12 | 16 |
| MLP Ratio | 4 | 4 |
| QKV bias | False | False |
| MLP bias | False | False |
| Droppath Rate | 0.0 | 0.1 |
| Layernorm eps | 1e-6 | 1e-6 |
| Initializer range | 0.02 | 0.02 |
| Use Calss Token | False | False |

Table 10: Detail config of ViT Tokenizer.

## F DiT Training and Inference Details

| Parameter | Value |
| --- | --- |
| Traing data | ImageNet |
| Training steps | 400k |
| Global Batch size | 256 |
| Optimizer | AdamW |
| Learning rate | $1.0 \times 10^{-4}$ |
| LR scheduler | Constant |
| beta1 | 0.9 |
| beta2 | 0.999 |
| weight decay | 0.0 |

Table 11: Training settings for DiT generation model.

Our primary experiments utilize DiT-XL for training generative models, with detailed parameters illustrated in the accompanying figure. The DiT-XL model requires approximately 40 hours of training on one H20 node to complete 400k steps.

For generative performance evaluation, we perform inference using DiT-XL without classifier-free guidance (CFG). During visualization, we employ a randomly selected CFG scale of 4.2.

