# OpenReview forum: "HieraTok: Multi-Scale Visual Tokenizer Improves Image Reconstruction and Generation"
_ICLR.cc/2026/Conference — ICLR 2026 Conference Withdrawn Submission_

### Official Review · Reviewer_Nrzb · 2025-10-29

**Soundness:** 2
**Presentation:** 2
**Contribution:** 2
**Rating:** 4
**Confidence:** 4

**Summary:**

This paper presents HieraTok, a multi-scale ViT-based image tokenizer for reconstruction and generation. HieraTok applies multi-scale down-sampling to the encoder features, and then uses a ViT decoder to reconstruct RGB images of the corresponding resolutions for training. By introducing scale-causal attention, HieraTok achieves performance improvements over single-scale tokenizers in both reconstruction and generation tasks, and conducts detailed ablation studies on design choices.

**Strengths:**

- The paper is generally well written and structured, making it easy to follow.
- The proposed HieraTok offers a solution to address the challenges of multi-scale image tokenization. By downsampling the encoder features and reconstructing corresponding RGB images, HieraTok improvements both image reconstruction and generation.
- Ablation results are detailed and show reasonable analysis of different model components.

**Weaknesses:**

- **Unfair comparison**: Since the decoder requires more input tokens, the comparison with the single-scale baseline (Table 7) is not fair. The paper lacks a truly fair comparison (e.g., upsampling single-scale tokens before feeding them into the decoder, adding learnable tokens to the single-scale decoder, or increasing the decoder size to maintain consistent FLOPs), which substantially weakens the persuasiveness of the reported results.
- **Incremental  improvement**: There are some concerns regarding the authors’ claim about the coarse-to-fine mechanism in the decoder. Since decoding is performed in a one-pass forward manner, for `Scale-independent Attention`, tokens of different scales merely share an FFN—there is no actual “progressive flow of information from coarse-grained semantic tokens to fine-grained structural tokens,” but rather just coarse-grained and fine-grained tokens. Moreover, the performance improvement of `Scale-causal Attention` over `Scale-independent Attention` is quite limited (Table 3), and the paper lacks a deeper analysis to substantiate the effectiveness of the coarse-to-fine design.

**Questions:**

- How does the fine-to-coarse version of `Scale-causal Attention` perform?
- Given the use of multi-scale features and *Scale-causal Attention*, why doesn’t the generator adopt VAR?
- More fair comparisons and detailed analyses are needed.

---

### Official Review · Reviewer_tMEC · 2025-10-30

**Soundness:** 3
**Presentation:** 3
**Contribution:** 2
**Rating:** 4
**Confidence:** 3

**Summary:**

This paper proposes HieraTok, a novel multi-scale Vision Transformer (ViT)-based visual tokenizer that overcomes the single-scale limitation of existing ViT tokenizers. Extensive experiments on ImageNet-256 show that HieraTok consistently improves both reconstruction and generation performance over single-scale ViT tokenizers.

**Strengths:**

1. The paper identifies a real limitation of ViT tokenizers—fixed-scale representation—and demonstrates a well-motivated architectural solution with measurable benefits.
2. The proposed multi-scale downsampling and scale-causal attention are conceptually straightforward and easily integrable into existing ViT architectures without changing token counts in downstream generative models.
3. The paper provides thorough ablations covering downsampling strategies (convolution vs. interpolation), attention types (scale-causal, scale-independent, full), and number of scales. The latent-space uniformity analysis (via t-SNE, entropy, and Gini coefficients) convincingly supports the design claims.

**Weaknesses:**

1. While effective, the idea of multi-scale representations has been extensively explored in FPN, Swin, and VAR-like architectures. The novelty mainly lies in adapting these ideas to ViT tokenizers rather than introducing a fundamentally new principle.
2. It is unclear why scale-causal attention is needed, in other words, whether it is necessary to restrict high-resolution tokens to interact exclusively with preceding low-resolution tokens.
3. Introducing multi-scale design will inevitably bring about more computation overhead, therefore, it would better to benchmark the GFLOPs/inference speed in Table 2.

**Questions:**

Please refer to the weakness part.

---

### Official Review · Reviewer_RnQM · 2025-10-30

**Soundness:** 2
**Presentation:** 2
**Contribution:** 2
**Rating:** 2
**Confidence:** 4

**Summary:**

This paper presents HieraTok, a novel multi-scale Vision Transformer tokenizer, and conducts experiments demonstrating that the hierarchical tokenizer indeed improves model performance.

**Strengths:**

- The paper is well-written and easy to follow.

 - The experimental results demonstrate that a hierarchical tokenizer can effectively improve performance.

**Weaknesses:**

- The comparison in Table 1 is unfair. Most methods compared under the ViT-based tokenizer setting are discrete tokenizers, whereas HieraTok is a continuous tokenizer and employs a DiT model as the generative backbone. Please compare HieraTok with other continuous tokenizers, or alternatively, provide results of a discrete version of HieraTok for fair comparison.

 - The improvement of the VAE through multi-scale design is rather trivial. While the multi-scale approach can indeed enhance reconstruction quality, a more complex VAE also makes generative model training significantly harder. It would be better to report the number of parameters for each model in Table 1 to clarify the trade-off between complexity and performance.

 - The results in Table 6 suggest that, when using generative models with the same number of parameters, the performance of HieraTok is much worse than that of even discrete tokenizers. Please verify this observation or provide an explanation for the discrepancy.

**Questions:**

NA

---

### Official Review · Reviewer_qU1U · 2025-11-02

**Soundness:** 2
**Presentation:** 2
**Contribution:** 2
**Rating:** 2
**Confidence:** 4

**Summary:**

**Problem.** ViT-based visual tokenizers typically learn single-scale latents, which may cap reconstruction fidelity and slow downstream generator convergence. **Method.** *HieraTok* introduces (i) **multi-scale token construction** by downsampling the encoder’s token map into a pyramid, concatenated low→high resolution, and (ii) a **scale-causal attention** in the decoder so high-res tokens attend to current+coarser scales only; plus **multi-scale RGB supervision** enforcing the approximation \(D(D_s(Z))\approx D_s(D(Z))\). **Key innovations.** Concatenate multi-scale token maps (Eq. 5), scale-causal attention (Eq. 7, Fig. 3), and scale-consistent reconstruction losses (Eqs. 8–9). **Main results.** On ImageNet-256, multi-scale vs single-scale improves rFID by 27.2% (1.47→1.07) and gFID by 18.9% (16.4→13.3) under matched training; scaled-up training reports rFID 0.45 and gFID 1.82 among ViT tokenizers (Tables 1–2, Fig. 4). **Significance.** If the gains are not confounded by capacity/compute, a multi-scale ViT tokenizer could be a stronger default for both reconstruction and generation.

**Strengths:**

- **Clear architectural proposal.** Multi-scale tokens via simple downsampling + a principled scale-causal attention (Fig. 2–3; Eqs. 5–7).
- **Consistent improvements under a fixed protocol.** Tables 2 and 6 show multi-scale wins across KL weights and DiT sizes.
- **Useful ablations.** Downsampling (interp vs conv) and attention (full/scale-independent/scale-causal) dissect where gains come from (Table 3).

**Weaknesses:**

1) **Capacity and compute confound the core claim.**
Your “multi-scale” model has **more parameters and FLOPs** and decodes **more tokens** (401M vs 346M; 230.45 vs 182.36 GFLOPs; 341 vs 256 decoder tokens; Table 7). Without parameter-, token-, and FLOP-matched controls, the 27.2% rFID and 18.9% gFID gains may largely reflect capacity. Please add: (a) a single-scale model with matched params/decoder tokens, (b) a multi-scale model trimmed to match single-scale FLOPs/tokens, and (c) report wall-clock per-step time. (Table 7).

2) **No end-to-end runtime/latency evidence.**
You claim faster convergence (Fig. 4), but report *steps-to-target* rather than **images/s** or **latency**. Please provide throughput, memory, and p95/p99 step time for tokenizer pretraining and DiT training; otherwise “1.38× speedup” is ambiguous. (Fig. 4).

3) **Novelty over multi-scale precedents is under-positioned.**
The design echoes FPN/U-Net top-down refinement (Fig. 3; §2.3). The paper claims “first multi-scale ViT tokenizer,” but related ViT tokenizers (FlexTok, TiTok, GigaTok, MAETok, ViTok) are listed (Table 1) without a **controlled head-to-head at equal compute**. Please: (i) compare against at least two ViT tokenizers with matched params/FLOPs; (ii) add a single-scale **decoder-only pyramid** baseline to test if the benefit is from supervision rather than tokens. (Table 1; §2.1–2.3).

4) **Scale-causal attention complexity and benefit are unclear.**
Scale-causal expands keys/values to all coarser levels (Eq. 7), potentially **increasing attention cost** over the concatenated sequence. Please report attention FLOPs/heads and compare to full attention and scale-independent in wall-clock **and** quality. (Eq. 7; Table 3).

5) **Multi-scale supervision may be doing the heavy lifting.**
The Appendix derivation \(D(D_s(Z))\approx D_s(D(Z))\) is assumed (App. B), but its empirical error is never measured. Provide \(\|D(D_s(Z))-D_s(D(Z))\|_2\) vs training time, and an ablation **without** multi-scale RGB supervision to quantify its standalone effect. (Eqs. 8–9; App. B).

6) **Statistics and robustness are thin.**
Core tables lack μ±σ across seeds; improvements like gFID 16.4→13.3 and rFID 1.47→1.07 could be sensitive. Please add ≥3 seeds, CIs, and **paired tests**, and report FID estimators’ CI. (Tables 2, 6; Fig. 4).

7) **Latent-space “uniformity” analysis is largely descriptive.**
t-SNE plots (Fig. 5) and Gini/entropy (Table 5) correlate with gFID but do not establish causality. Please follow VAVAE-style protocols with stronger geometry diagnostics and show how these metrics predict *per-class* convergence, not just aggregates. (Fig. 5; Table 5).

8) **Downsampling choice is under-analyzed for generation.**
You conclude **conv downsampling** helps generation while interpolation does not (Table 3). Why? Provide spectra/feature-rank analyses of \(Z_s\), and visualize cross-scale attention patterns to show semantics-→detail flow. (Table 3).

9) **Limited scope: single dataset/resolution and one paradigm.**
Experiments are ImageNet-256 only; no COCO/LAION subsets, no higher resolution, and only DiT as the generator. Please add at least one **additional dataset** and **another generator family** (e.g., MAR/AR or LDM) to establish generality. (§5).

10) **Reproducibility & fairness.**
Implementation details exist (App. E) but not full configs/seeds/checkpoints. Also, “training fairness” is argued by matching epochs, but equal **epochs** ≠ equal **compute** when sequences/params differ. Release scripts and report matched-compute comparisons. (App. E; C.2).

**Questions:**

See Weaknesses

---

### Note · Authors · 2025-11-12

I have read and agree with the venue's withdrawal policy on behalf of myself and my co-authors.